# The role of structured reporting and structured operation planning in functional endoscopic sinus surgery

**Benjamin Philipp Ernst**[1]*, **Manuel René Reissig**[1], **Sebastian Strieth**[2], **Jonas Eckrich**[1], **Jan H. Hagemann**[1], **Julia Döge**[1], **Christoph Matthias**[1], **Haralampos Gouveris**[1], **Johannes Rübenthaler**[3], **Roxanne Weiss**[4], **Wieland H. Sommer**[3], **Dominik Nörenberg**[5], **Thomas Huber**[5], **Phillipp Gonser**[6], **Sven Becker**[6], **Matthias F. Froelich**[5]

1 Department of Otorhinolaryngology, University Medical Center Mainz, Mainz, Rhineland-Palatinate, Germany, 2 Department of Otorhinolaryngology, University Hospital Bonn, Bonn, North Rhine-Westphalia, Germany, 3 Department of Radiology, LMU University Hospital, Munich, Bavaria, Germany, 4 Department of Otorhinolaryngology, University Hospital Frankfurt, Frankfurt, Hessen, Germany, 5 Department of Radiology and Nuclear Medicine, University Medical Center Mannheim, Mannheim, Baden-Wuerttemberg, Germany, 6 Department of Otorhinolaryngology, Head and Neck Surgery, University of Tübingen Medical Center, Tübingen, Baden-Wuerttemberg, Germany

* benjamin.ernst@unimedizin-mainz.de

**Data Availability Statement:** All relevant data are within the manuscript and its Supporting information files.

## Abstract

Computed tomography (CT) scans represent the gold standard in the planning of functional endoscopic sinus surgeries (FESS). Yet, radiologists and otolaryngologists have different perspectives on these scans. In general, residents often struggle with aspects involved in both reporting and operation planning. The aim of this study was to compare the completeness of structured reports (SR) of preoperative CT images and structured operation planning (SOP) to conventional reports (CR) and conventional operation planning (COP) to potentially improve future treatment decisions on an individual level. In total, 30 preoperative CT scans obtained for surgical planning of patients scheduled for FESS were evaluated using SR and CR by radiology residents. Subsequently, otolaryngology residents performed a COP using free texts and a SOP using a specific template. All radiology reports and operation plannings were evaluated by two experienced FESS surgeons regarding their completeness for surgical planning. User satisfaction of otolaryngology residents was assessed by using visual analogue scales. Overall radiology report completeness was significantly higher using SRs regarding surgically important structures compared to CRs (84.4 vs. 22.0%, p<0.001). SOPs produced significantly higher completeness ratings (97% vs. 39.4%, p<0.001) regarding pathologies and anatomical variances. Moreover, time efficiency was not significantly impaired by implementation of SR (148 s vs. 160 s, p = 0.61) and user satisfaction was significantly higher for SOP (VAS 8.1 vs. 4.1, p<0.001). Implementation of SR and SOP results in a significantly increased completeness of radiology reports and operation planning for FESS. Consequently, the combination of both facilitates surgical planning and may decrease potential risks during FESS.

**Funding:** The authors received no specific funding for this work.

**Competing interests:** Wieland H Sommer is the founder and CEO of the company Smart Reporting GmbH, which hosts an online platform for structured reporting. Dominik Nörenberg and Thomas Huber are part-time employees of Smart Reporting GmbH. Matthias F Froelich is a medical consultant of Smart Reporting GmbH. The other authors of this manuscript declare no relationships with any companies, whose products or services may be related to the subject matter of the article. This manuscript is part of a medical doctoral thesis presented by Manuel R Reissig at the University Mainz Medical School. This does not alter our adherence to PLOS ONE policies on sharing data and materials.

## Introduction

Functional endoscopic sinus surgery (FESS) represents the gold standard in the surgical management of paranasal sinus disease [1, 2]. Three-dimensional high-resolution computed tomography (CT) scans are required for any surgical approach to treat both chronic and acute rhinosinusitis, particularly with potentially occurring complications [3, 4]. Prior to surgery, CT scans are indispensable to determine the extent of FESS as well as to identify anatomical structures with an increased risk regarding catastrophic complications. Such complications may include injuries of the anterior skull base, the orbit, optic nerve or internal carotid artery with potential lethal bleedings, ischemic stroke or need for blood transfusions [5–9]. Especially in the era of powered instruments such as microdebriders, a profound understanding of the anatomy is of utmost importance, as these particular instruments can cause severe complications when handled without sufficient care [10, 11]. Therefore, a solid preoperative knowledge of the anatomy using radiological and surgical knowledge is key for any successful interdisciplinary surgical treatment planning which may prevent premature revision surgery [12]. Additionally, the accessibility of CT scans sets the foundation towards performing image-guided surgery by using intraoperative navigation [13].

Despite major improvements in the quality of CT scans making it possible to address a wide variety of diseases and anatomical variances in great detail, radiologists and otolaryngologists have grossly different perspectives on these scans [4, 14]. Whereas radiologists mainly focus on the extent and the effects of the disease, otolaryngologists are also very much interested in anatomical variances to plan surgical approaches [15]. In addition, residents in training, both in radiology and otolaryngology, often struggle which anatomical structures which have to be considered in reporting and operation planning and how to grade them [16, 17]. Conventional CT checklists have been implemented in many otolaryngology departments to give residents some sort of guidance. Nevertheless, these checklists usually exist in analogous form. Therefore, they are frequently abandoned after initial implementation as they are not directly integrated into the clinical workflow [18].

Structured reporting (SR) has been advocated for various diagnostic modalities in radiology as well as in otolaryngology and other specialties with impact on treatment decisions and interdisciplinary communication [19–25]. The main benefit, especially for residents in training, is the standardization of the content and terminology which is known to improve report quality and time efficiency when compared to conventional reporting (CR) [19–21]. Additionally, the implementation of structured operation planning potentially reduces the risk of missing key structures during the planning process. This may be due to the fact that structured reporting templates can highlight important features as well as pertinent negative findings within predefined checklists [26, 27]. Consequently, SR and SOP may also promote the learning curve of younger radiologists and otolaryngologists in training [28].

Overall, the aim of this study was to perform a comparative analysis of the completeness of CT reports and surgical plannings using SR and SOP, respectively, to the conventional approaches (e.g. CR and COP) which may have the potential to improve treatment decisions and patient outcome in the long term.

## Materials and methods

### Study design and study sample

This study was designed to evaluate the impact of structured reporting as well as structured operation planning in the process of the surgical management of chronic rhinosinusitis. Therefore, n = 30 consecutive preoperative CT scans of adult patients (m = 17, w = 13, mean

**Table 1. Demographics and characteristics of the study sample.**

| Characteristics | Value |
|---|---|
| Number of patients included | 30 |
| Age at surgery (mean ± SD) | 41.4 ± 12.8 years (range: 23–58 years) |
| Gender | Male: n = 17 Female: n = 13 |
| Indication for functional endoscopic sinus surgery | Chronic rhinosinusitis with polyposis: n = 20 |
| | Chronic rhinosinusitis without polyposis: n = 10 |
| Number of participating otolaryngology residents | 6 |
| Years of residency | 4.5 ± 0.9 (range: 4–6 years) |

age at surgery: 38.2 years) scheduled for FESS due to chronic rhinosinusitis with (n = 10) and without polyposis (n = 20) were included in this study (see Table 1). Existing conventional reports of sinus CTs were identified. These CT scans were assessed once again using a specifically designed online-based SR template (n = 30, Smart Reporting GmbH, Munich, Germany, https://smart-reporting.com) by one board-certified radiologist. The template was designed in consideration of the latest radiological and otolaryngological recommendations for preoperative CT analysis before FESS procedures, anatomical structures and terminology. The templates guide the user through clickable decision-trees specific for a wide range of benign and malignant paranasal sinus diseases. By completing the decision-trees, the software creates full semantic sentences from previously defined text elements, thus creating complete and consistent reports. In order to reduce the likelihood to consult additional medical literature or classification tables during reporting and planning, info boxes can be added to the template displaying classifications and medical guidelines which have been shown to improve time efficiency [19].

Subsequently, residents in training (n = 6, 4.5 ± 0.9 years of residency) who were assigned to the particular FESS procedure were to plan the operation by conventional operation planning (COP, n = 30) using the CT scans on file. To do so, they completed the department's standard FESS planning form in which the categories nasal septum, middle nasal meatus, ethmoid infundibulum, maxillary sinus, ethmoid sinus, sphenoid sinus and frontal sinus have to reported upon for each side in writing by hand. In a second step, residents planned the same procedures using a specifically designed online-based template for SOP (n = 30) of FESS procedures (Smart Reporting GmbH, Munich, Germany, https://smart-reporting.com). Time needed for each type of planning was recorded.

Finally, participating residents completed a user satisfaction questionnaire regarding COP and SOP, respectively.

## Ethics approval and consent to participate

Ethics approval was obtained by the Institutional Review Board (Ethik-Kommission der Landesärztekammer Rheinland-Pfalz. Reference number: 2018–13225). All procedures performed in studies involving human participants were in accordance with the ethical standards of the institutional and national research committee and with the 1964 Helsinki declaration and its later amendments or comparable ethical standards.

Oral and written patient information was given by the examining physician. Written informed consent was obtained prior to the examination.

## Sample size calculation

As previously described in the literature, the number of patients needed was calculated based on the anticipated effect size when comparing the percentage of CRs/COPs with 80%

completeness or higher to SRs/SOPs. We assumed that 45% of CRs/COPs would have very high completeness ratings (i.e. of 80% or higher), considering the report completeness of other head and neck ultrasound imaging studies as shown in the literature [19–21]. Additionally, we estimated that completeness ratings would go up to 80% by using SRs/SOPs. The power was set to 80% with a significance level of α = 0.05. Using these parameters, the minimum number of reports required for the study was calculated to be n = 58 (29 reports in each group).

## Evaluation of radiological reports and operation plannings

Collectively, 60 anonymized reports (30 SRs and CRs each) were independently analyzed for completeness (i.e. reporting of critical anatomy and pathology) by a board-certified radiologist and an otolaryngologist. Additionally, an evaluation template was implemented in order to standardize the analysis.

Following this, 60 anonymized operation plannings (30 SOPs and COPs each) were also evaluated for completeness (i.e. identification of critical anatomical structures and disease as well as determination of the extent of the procedure) by two board-certified otolaryngologists with a high expertise in FESS based on an evaluation checklist. Due to the free text structure of COPs which contain handwritten plannings, readability was subjectively assessed using a five-point Likert scale. Time expenditure for each type of planning was also recorded and compared.

Otolaryngology residents were asked for their opinions regarding practicability (question 1), usefulness in everyday practice (question 2), improvement in report-quality (question 3), time efficiency (question 4), justification of additional time needed (question 5) as well as benefits for inexperienced physicians being trained in FESS (question 6) of both types of planning by applying a ten-point visual analogue scale.

## Statistical analysis

Data are reported as the mean percentage of the greatest possible outcome (i.e. percentage of maximum completeness, etc.), mean time needed to complete the report (seconds) and mean VAS values ± SD. Wilcoxon signed-rank test for paired nominal data was used to compare operation planning evaluations and questionnaire findings while Mann-Whitney test for unpaired data was used to compare radiology reports. A p-value of less than 0.05 was considered statistically significant. All statistical analyses were performed using Prism 8 (GraphPad Software, Inc., San Diego, CA, USA).

# Results

## Analysis of radiological reports

Radiological reports, both CR and SR, were completed by radiology residents. Compared to CRs, SRs showed a significantly higher overall completeness level for all categories (84.4 vs. 22.0%, p<0.001, Fig 1a).

In particular, SRs were more complete in regard to description of nasal septum (100 vs. 0%, p<0.001), middle nasal meatus (85.8 vs. 3.3%, p<0.001), maxillary sinus (95.6 vs. 62.2%, p<0.001), ethmoid sinus (82.7 vs. 24.3%, p<0.001), and frontal sinus (75 vs. 33.3%, p<0.001). Additionally, the anterior skull base (98.3 vs. 1.7%, p<0.001) and the absence of potential masses (70 vs. 13.3%, p<0.001) were reported more frequently. Concerning classifications, SRs had a higher level of completeness in regard to the Keros classification (83.3 vs. 0%, p<0.001) and the Lund-Mackay Score (73.3 vs 0%, p<0.001). These results are summarized in Fig 1b.

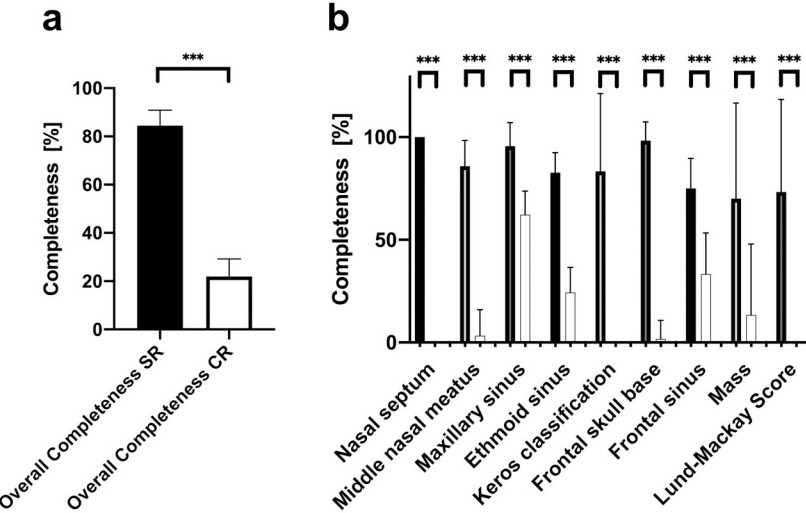

**Fig 1. Comparison of overall completeness between structured reports (SR) and conventional reports (CR) of paranasal sinus computed tomographies (CT).** Analysis reveals significantly superior completeness ratings of SR (a). Detailed completeness assessment of relevant items reveals superior results of SR in all items when compared to CR (b). *** $p < 0.001$.

## Analysis of operation plannings

Surgical planning of FESS, by both COP and SOP, was performed by otolaryngology residents who were assigned to assist or to perform (under supervision of a board-certified otolaryngologist) each FESS procedure by using the preoperative CT scans for surgical planning. When compared to mostly handwritten COPs, SOPs showed a significantly better readability (100 vs. 62%, p<0.001). Analysis revealed significantly higher overall completeness ratings for SOP when compared to COP (97 vs. 39.4%, p<0.001). In detail, SOPs received significantly superior ratings in respect to the nasal septum (100 vs. 56.7%, p<0.001), the middle nasal meatus (96.7 vs. 54.2%, p<0.001), the ethmoid infundibulum (92.5 vs. 50%, p<0.001) and the maxillary sinus (93.3 vs. 44.7%, p<0.001). In addition, the ethmoid sinus (99.2 vs. 33.3%, the sphenoid sinus (93.9 vs. 30%, p<0.001) and the frontal sinus (98.6 vs. 31.9%, p<0.001) were documented more completely using SOP. Potential masses in the paranasal sinuses (100 vs. 15%, p<0.001) and particularly the Keros classification of the anterior skull base (100 vs. 40%, p<0.001) were considered significantly more reliably using SOP as well. Evaluation of time needed to plan the operation shows a tendency towards a better time efficiency for SOPs without reaching significance level (148 s vs. 160 s, p = 0.61). A comprehensive presentation of operation planning analysis is shown in Fig 2.

## User satisfaction

In total, the questionnaire showed a significant preference for SOPs by all participating otolaryngology residents (8.1 vs. 4.1, p<0.001). In detail, the use of SOP was rated to be usable in everyday practice (7.8 vs. 3.7, p = 0.035), to improve the quality of preoperative planning (8.8 vs. 4.0, p = 0.038) and to have a favorable time efficiency (7.2 vs. 3.5, p = 0.04). Additionally, residents thought that any extra time spent on SOP was justified compared to COP (8.2 vs. 3.3, p = 0.038). All other questions revealed a positive tendency towards a preference of SOP without reaching the level of significance. A detailed analysis of questionnaires is shown in Fig 3.

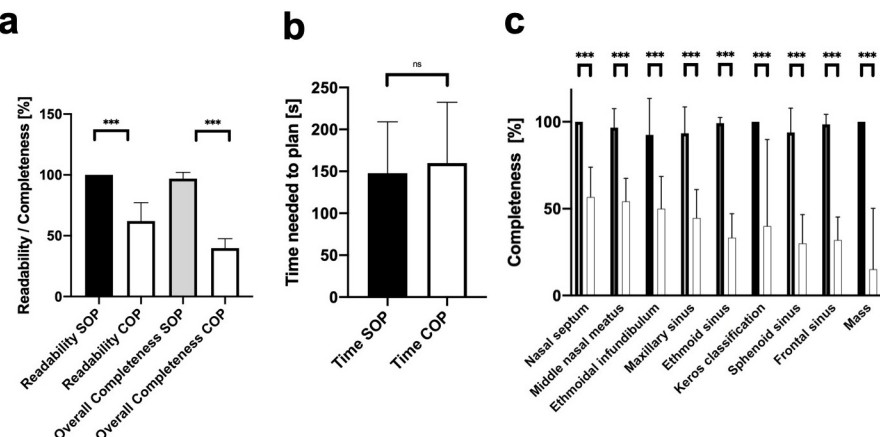

**Fig 2. Analysis of readability, time efficiency and completeness of structured and conventional operation plannings.** Structured operation plannings (SOP) reveal a significantly superior readability and overall completeness when compared to conventional operation planning (COP, a). Evaluation of time needed to plan the operation shows a tendency towards a better time efficiency for SOPs without reaching significance level (b). Analysis of detailed completeness levels of relevant anatomical features for FESS plannings outlines significantly higher completeness ratings for all analyzed items when compared to COP (c). n.s. = not significant, *** $p < 0.001$.

## Discussion

FESS has been proven as an effective surgical treatment of refractory chronic conditions of paranasal sinuses and, therefore, represents the surgical gold standard in these cases [29]. While technical developments that lead to modern FESS procedures have improved the surgical efficacy even further [30–32], the influence of structured approaches in radiology reports and pre-treatment surgical plannings have not been studied comprehensively. Yet, precise imaging and accurate reporting of findings are generally found to be relevant for therapy

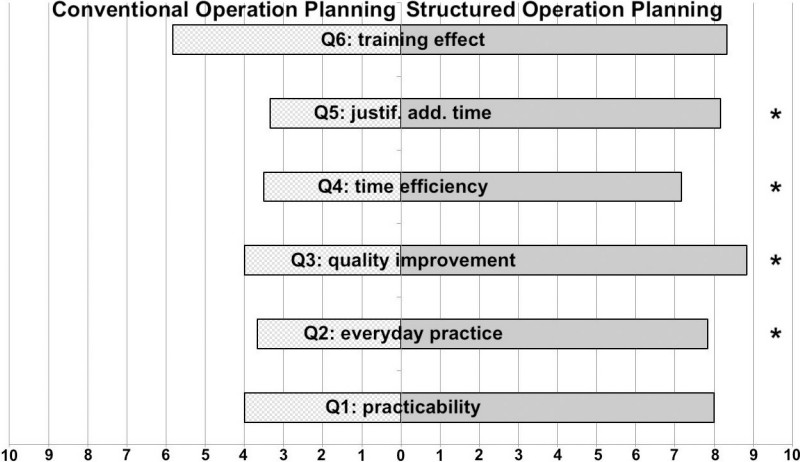

**Fig 3. Analysis of user satisfaction.** User satisfaction analysis reveals a significantly superior overall satisfaction when using structured operation planning (SOP) when compared to conventional operation planning (COP). In detail, SOP received significantly higher ratings concerning everyday practice (Q2), quality improvement (Q3) and time efficiency (Q4). In case of additional time needed for SOP, users thought that this additional time was likely to be well spent (Q5). The items practicability (Q1) and training effect (Q6) showed a tendency toward better ratings for SOP without reaching significance level. * $p < 0.05$.

planning. Furthermore, detailed planning of surgery can increase confidence, boost the learning curves of residents in training and reduce complications during and after surgery [33, 34].

The main finding of our study outlines an added value of a structured approach to radiology reporting in combination with structured surgical planning to improve treatment decisions. In summary, structured data acquisition did indeed lead to superior completeness, both for information collected by otolaryngology and radiology residents. This may very well be due to the fact that radiologists and otolaryngologists often focus on quite different aspects when evaluating CT scans of the paranasal sinuses [15, 35, 36]. Whereas radiologists mainly evaluate CT scans for potential pathologies and their consecutive extent (e.g. signs and extent of chronic rhinosinusitis), otolaryngologists usually reflect CT scans in terms of a surgical approach considering the pathological findings [37]. Interestingly, implementation of SOP caused otolaryngology residents to consider potentially dangerous structures such as the lamina papyracea and particularly the anterior skull base and its asymmetries as described by the Keros classification (100 vs. 31.3%, p = 0.0007) significantly more often. While these results clearly demonstrate superior completeness ratings of SOPs, the insufficient completeness of COPs may be due to the fact that the SOP template queries all structures for every operation planning while unremarkable findings may not be reported using COPs due to a lack of necessity. Consequently, while important structures are actually considered by otolaryngology residents for each and every surgery, this may be underrepresented in COP documentation. Nevertheless, without proper documentation, the chance to miss an atypical critical structure in the surgical planning process, which may lead to intraoperative complications, is increased. Additionally, the surgical approach to the frontal sinus was described significantly more frequent which may very well reduce the rate of revision surgery due to insufficient drainage of the frontal sinus [38, 39]. As described by Stammberger et al., the uncapping the egg technique to the frontal recess and sinuses requires thorough preparation in order to identify ethmoidal cells protruding into the frontal sinus [40]. If planned insufficiently, dissection of only the lower portion of such cells will not provide sufficient drainage of the frontal sinus and consequently will fail to resolve the frontal sinus disease.

Structure and content of radiology reports of paranasal sinuses in terms of surgical therapies may be unclear to radiologists, and in particular for inexperienced residents, even though these scans are performed as part of the surgical planning [15, 35, 36]. Therefore, the creation of specific structured reporting templates may also enhance interdisciplinary discussion and may provide common grounds for exchange transmission of information [4]. The template used in this study was carefully designed in consideration of the current guidelines by a team of board-certified otolaryngologists and radiologists with a special expertise in FESS and head and neck radiology, respectively. In this process, anatomical structures that are of central importance for radiological reporting and FESS were identified and their interdisciplinary importance was discussed to implement all necessary information for detailed and accurate surgical planning. In this course, an interdisciplinary approach to preoperative reporting and planning of FESS was established. This may be highly desirable not only from a surgical perspective, but also from radiologists' point of view in order to maximize the clinical value of their reports leading to higher satisfaction of referring physicians. Additionally, it decreases the risk that atypical anatomical structures, which may be concealed by extensive paranasal sinus disease, are missed, both by radiologists and otolaryngologists, and therefore compromised during surgery [41–43].

These results are in line with previously reported findings on structured reporting for imaging: In radiology, structured reporting has shown superior results in terms of report completeness, quality and time efficiency when compared to conventional free-text reporting [22, 44]. Therefore, structured reporting is recommended by leading societies [45, 46]. Additionally,

the utilization of structured reporting for pathology reports did lead to better patient management and outcomes [47] and showed beneficial effects in the setting of ENT sonography reporting [19–21]. While several studies report higher completeness, better readability, advantages in information extraction and educational benefits [19–24, 44, 48–50], sometimes a too rigid structure is claimed to be an obstacle for experienced readers in diagnostic imaging [51]. However, the results presented in this study show a comparable time investment towards surgical planning regarding both approaches. Due to the fact that the pre-existing CRs that had been created during the daily clinical practice, no information was available regarding the time needed to finish the report. Such being the case, no statement can be made concerning differences in time efficiency between CR/SR of paranasal sinus CTs. Also, structured reporting templates are now offering more flexible solutions and allowing the utilization of pre-defined text elements that can help to reduce orthographic mistakes as they are based on expert-reviewed text components [45]. This may also enhance report quality in the era of telemedical approaches which often suffer from non-native speaking reporting physicians [52–54]. Additionally, most of the recent publications have dismissed the problem of impaired radiology reports due to too rigorous structures which may be attributed to enhanced information technologies incorporated into SR templates [19–21].

The results presented in this study have to be interpreted in the context of its design: First, the results of this study are based on treatment planning for only one surgical procedure, namely FESS. As the surgical approach varies noticeably depending on the pathological pattern and the department's internal standard and expertise, structured therapy planning may be of a very high importance in order to increase standardization. This may secondly lead to an increased scientific comparability of radiological findings as well as surgical planning, especially in the context of multicenter studies [55, 56]. Therefore, generalization of the results to other surgical procedures should only be done with great care. Additionally, the operation planning was carried out by residents in training, which does not represent the standard of care of board-certification. Second, imaging reporting of paranasal sinuses is a very suitable application for structured reporting in comparison to other diagnostic procedures with a higher degree of variability of pathological findings, like for example an MRI scan for unclear vertigo. In addition, structures to be reported may be valued significantly different by radiologists and otolaryngologists. In consequence, SR may be a valuable tool to ensure standardization while enabling the necessary variability. Third, while implementation of SR/SOP resulted in a significant increase of completeness, this study did not evaluate correctness of the reports. Therefore, no statement concerning the impact of SR/SOP on report content accuracy can be made. This has to be evaluated in future studies. Fourth, since participating Otolaryngology residents used the same CT scans for corresponding SOPs and COPs, potential bias due to testing or learning effects cannot be ruled out. A sequence of creating COPs before SOPs was chosen in order to reduce bias since, unlike SOPs, COPs do not offer any feedback to the user. Consequently, potential training effects are minimized to greatest possible extent. Additionally, residents prepared SOPs and COPs before the operation to reduce potential bias that may arise from additional knowledge acquired from intraoperative findings. Since there were only n = 6 residents with a sufficient experience in FESS procedures available at the University Medical Center Mainz, each resident prepared 5 corresponding SOPs/COPs within this study. By assigning residents one after the other, potential bias due to learning effects cannot be ruled out but are minimized since a greatest possible time interval between assignments may blur details used within SOPs. Finally, further analyses in other patient collectives, other healthcare systems and other treatment modalities may be needed to validate the findings presented in this study.

## Conclusions

Implementation of SR and SOP results in a significantly increased completeness of radiologic reports and operation planning in the surgical management of paranasal sinus disease. Consequently, the combination of both may enhance the learning curve, both in radiology and otolaryngology, and decrease potential risks during endoscopic sinus surgery in rhinology centers.

## Supporting information

**S1 Fig. Flowchart of the structured radiology reporting and structured operation planning structures.**
(TIF)

**S2 Fig. Screenshot of structured reporting template.** For full decision tree, refer to S1 Table.
(TIF)

**S3 Fig. Screenshot of structured operation planning template.** For full decision tree, refer to S1 Table.
(TIF)

**S1 Table. Key features of the structured reporting template.**
(DOCX)

**S2 Table. Key features of the structured operation planning template.**
(DOCX)

**S3 Table. Supporting raw data evaluated in this study.**
(XLSX)

## Acknowledgments

The authors thank Ben Braun for language editing.

## Author Contributions

**Conceptualization:** Benjamin Philipp Ernst, Sebastian Strieth, Wieland H. Sommer, Sven Becker, Matthias F. Froelich.

**Data curation:** Benjamin Philipp Ernst, Manuel René Reissig, Jonas Eckrich, Matthias F. Froelich.

**Formal analysis:** Benjamin Philipp Ernst, Manuel René Reissig, Jonas Eckrich, Matthias F. Froelich.

**Investigation:** Benjamin Philipp Ernst, Manuel René Reissig, Jonas Eckrich, Jan H. Hagemann, Julia Döge, Haralampos Gouveris, Johannes Rübenthaler, Roxanne Weiss, Dominik Nörenberg, Thomas Huber, Phillipp Gonser, Sven Becker, Matthias F. Froelich.

**Methodology:** Benjamin Philipp Ernst, Phillipp Gonser, Matthias F. Froelich.

**Project administration:** Benjamin Philipp Ernst, Sebastian Strieth, Matthias F. Froelich.

**Resources:** Benjamin Philipp Ernst.

**Software:** Benjamin Philipp Ernst, Wieland H. Sommer.

**Supervision:** Benjamin Philipp Ernst, Christoph Matthias, Matthias F. Froelich.

**Validation:** Benjamin Philipp Ernst, Matthias F. Froelich.

**Visualization:** Benjamin Philipp Ernst, Matthias F. Froelich.

**Writing – original draft:** Benjamin Philipp Ernst, Jonas Eckrich, Haralampos Gouveris, Johannes Rübenthaler, Roxanne Weiss, Dominik Nörenberg, Sven Becker, Matthias F. Froelich.

**Writing – review & editing:** Benjamin Philipp Ernst, Sebastian Strieth, Haralampos Gouveris, Johannes Rübenthaler, Dominik Nörenberg, Sven Becker, Matthias F. Froelich.

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
