## [Decision Letter · Decision Letter 0]

6 Oct 2020

PONE-D-20-22030

The role of structured reporting and structured operation planning in functional endoscopic sinus surgery

PLOS ONE

Dear Dr. Ernst,

Thank you for submitting your manuscript to PLOS ONE. After careful consideration, we feel that it has merit but does not fully meet PLOS ONE’s publication criteria as it currently stands. Therefore, we invite you to submit a revised version of the manuscript that addresses the points raised during the review process.

We look forward to receiving your revised manuscript.

Kind regards,

Giannicola Iannella, M.D

Academic Editor

PLOS ONE

Journal Requirements:

2. Thank you for providing the full details of your ethics board approval and informed patient consent in the Ethics Statement. We ask that you additionally include this information in the Methods section.

3.Thank you for stating the following in the Competing Interests section:

[Wieland H Sommer is the founder and CEO of the company Smart Reporting GmbH, which hosts an online platform for structured reporting. Dominik Nörenberg and Thomas Huber are part-time employees of Smart Reporting GmbH. Matthias F Froelich is a medical consultant of Smart Reporting GmbH. The other authors of this manuscript declare no relationships with any companies, whose products or services may be related to the subject matter of the article.  This manuscript is part of a medical doctoral thesis presented by Manuel R Reissig at the University Mainz Medical School.].

Reviewers' comments:

Reviewer's Responses to Questions

**Comments to the Author**

1. Is the manuscript technically sound, and do the data support the conclusions?

Reviewer #1: Yes

Reviewer #2: Partly

2. Has the statistical analysis been performed appropriately and rigorously? 

Reviewer #1: Yes

Reviewer #2: Yes

3. Have the authors made all data underlying the findings in their manuscript fully available?

Reviewer #1: Yes

Reviewer #2: No

4. Is the manuscript presented in an intelligible fashion and written in standard English?

Reviewer #1: Yes

Reviewer #2: Yes

5. Review Comments to the Author

Reviewer #1: In this original article, the authors compared the completeness of structured reports of preoperative CT images and structured operation planning to conventional reports and conventional operation planning to improve treatment decisions on an individual level in 30 patients undergoing FESS surgery. The paper is well written and the topic interesting. References are adequate. I recommend acceptance in the present form.

Reviewer #2: Potential for Conflict of Interest:

Because of the relationship of some authors with SmartReporting GmbH, provide more information about the statement “no relationships with any companies, whose products or services may be related…”. Even if there were no actual conflicts of interest, is there any management plan (e.g., by the University of Mainz) for any potential, possible, future conflict of interest?

Methods:

- Submit a flow-chart of the SR and SOP structures and decision-trees for review (as supporting information).

- How many residents participated in preparing report? Were they at the same stage in their training? Did a given resident do an SR and a CR for the same patient? If so, would this introduce a bias for which ever report was prepared secondly? There are similarly questions if a given resident prepared reports on more than one patient.

Results:

(1) Is there any data concerning an association or correlation between “completeness” (an outcome measure) and “treatment decisions on an individual level” (part of the overall aim of the study)?

Discussion:

(a) How could it be determined whether the use of SR (for pre-operative CT images) and SOP “may decrease potential risks during FESS?

(b) What about having a structured outcomes report (“SOR” for post-operative outcomes) to go along with the SR and SOP?

6. PLOS authors have the option to publish the peer review history of their article (what does this mean?). If published, this will include your full peer review and any attached files.

Reviewer #1: No

Reviewer #2: **Yes: **John H Anderson, MD, PhD

---

## [Author Response · Author response to Decision Letter 0]

11 Oct 2020

Dear Ladies and Gentlemen,

we would like to thank the reviewers and editor for giving our manuscript so much consideration. We carefully revised our manuscript PONE-D-20-22030 according to the reviewer’s suggestions and feel confident that the manuscript meets your expectations by now. 

Editor: 

and

The manuscript and the associated files meet the PLOS ONE’s style requirements.

2. Thank you for providing the full details of your ethics board approval and informed patient consent in the Ethics Statement. We ask that you additionally include this information in the Methods section.

The Ethics Statement was included in the Methods sections of our revised manuscript.

[Wieland H Sommer is the founder and CEO of the company Smart Reporting GmbH, which hosts an online platform for structured reporting. Dominik Nörenberg and Thomas Huber are part-time employees of Smart Reporting GmbH. Matthias F Froelich is a medical consultant of Smart Reporting GmbH. The other authors of this manuscript declare no relationships with any companies, whose products or services may be related to the subject matter of the article. This manuscript is part of a medical doctoral thesis presented by Manuel R Reissig at the University Mainz Medical School.].

Since there are no restrictions, the statement "This does not alter our adherence to PLOS ONE policies on sharing data and materials.” Was added to our Competing Interest statement in our cover letter.

4. Have the authors made all data underlying the findings in their manuscript fully available? (Reviewer #2: No)

As part of the PLOS ONE policy on data sharing, we prepared a supporting information table that provides the raw data that includes:

• The values behind the means, standard deviations and other measures reported;

• The values used to build graphs;

• The points extracted from images for analysis.

When revisiting our entire raw data collection to prepare the supporting information table (S3 Table), we noticed a calculation mistake concerning the mean VAS values of our COP user satisfaction questionnaire and have corrected it accordingly. Consequently, mean VAS values for COP as well as p-values for questions 1 (Q1) through question 6 (Q6) slightly differ. However, these changes did not result in any changes regarding the level of significance of inter-group-varability. Mean values for SOP as well as overall user satisfaction that represents the sum of all six questions was calculated correctly, both for SOP and COP. Changes can be tracked in the “tracked changes” version of our revised manuscript. The corrected values can be objectively reproduced through the raw data provided in the S3 Table. Figure 3 was revised accordingly.

Old values: New values:

User satisfaction Q1: 8.0 vs. 5.5, p>0.05 8.0 vs. 4.0, p=0.058

User satisfaction Q2: 7.8 vs. 5.3, p=0.041 7.8 vs. 3.7, p=0.035

User satisfaction Q3: 8.8 vs. 6.0, p=0.022 8.8 vs. 4.0, p=0.038

User satisfaction Q4: 7.2 vs. 5.3, p=0.028 7.2 vs. 3.5, p=0.04

User satisfaction Q5: 8.2 vs. 5.4, p=0.018 8.2 vs. 3.3, p=0.038

User satisfaction Q6: 8.3 vs. 6.7, p>0.05 8.3 vs. 5.8, p=0.34

Time efficiency: 148 vs. 160s, p=0.71 148 vs. 160s, p=0.61

We apologize sincerely for these miscalculations that were honest mistakes that did not happen in bad faith. We would like to take an entirely open approach in addressing this topic by disclosing it here. As mentioned above the level of significance of differences between groups was not changed by these recalculations and therefore there was no impact on the conclusions of this study.

Reviewer #1: 

In this original article, the authors compared the completeness of structured reports of preoperative CT images and structured operation planning to conventional reports and conventional operation planning to improve treatment decisions on an individual level in 30 patients undergoing FESS surgery. The paper is well written and the topic interesting. References are adequate. I recommend acceptance in the present form.

Thank you very much for your review and your recommendation for acceptance of our manuscript.

Reviewer #2: 

1. Because of the relationship of some authors with Smart Reporting GmbH, provide more information about the statement “no relationships with any companies, whose products or services may be related…”. Even if there were no actual conflicts of interest, is there any management plan (e.g., by the University of Mainz) for any potential, possible, future conflict of interest?

Thank you very much for your comment. As the reviewer pointed out correctly, Wieland H Sommer is the founder and CEO of Smart Reporting GmbH. Dominik Nörenberg and Thomas Huber are part-time employees of Smart Reporting GmbH. Matthias F Froelich is a medical consultant of Smart Reporting GmbH. The human resources department of University Medical Centre Mannheim has approved all part-time and consulting contracts. None of the other authors does currently have a relationship with Smart Reporting GmbH. 

In addition, University Medical Center Mainz has a management plan in place for handling of potential conflicts of interests for the institution and its employees including several standardized procedures and thorough approval processes by the management in cooperation with both the legal department and the human resources department:

• Every employment contract of employees of University Medical Center Mainz are subject to approval by the personnel department. The corresponding form can be found here: https://www.verwaltung.personal.uni-mainz.de/files/2020/06/Antrag-NT-Beamte-neu.pdf

Furthermore, a corresponding employment contract would have to be carried out without opposing the activities for University Medical Center Mainz. This is stated specifically in corresponding legislation, the “Landesgesetz über die Errichtung der Universitätsmedizin der Johannes Gutenberg-Universität Mainz“ in §10, §35 and §36: http://landesrecht.rlp.de/jportal/portal/t/y9j/page/bsrlpprod.psml?doc.hl=1&doc.id=jlr-JohGutUniMedGRPrahmen:juris-lr00&documentnumber=1&numberofresults=44&showdoccase=1&doc.part=X&paramfromHL=true

• There is no agreement or contract regarding the publication or other topics between Smart Reporting GmbH and the ENT department of University Medical Center Mainz. The publication process was initiated, maintained and supervised by the corresponding author at the lead site, University Medical Center Mainz. Every future contract would be subject of approval by the legal department of University Medical Center Mainz. Additionally, the institutional ethics committee audited potential conflicts of interest before giving approval for this study.

• Additionally, while the founder and employees of Smart Reporting GmbH were involved in the planning phase of this study and contributed by providing the structured reporting templates within the Smart Reporting Software Solution free of charge. Data were analyzed by employees of the University Medical Center Mainz with no relationship to Smart Reporting GmbH. Data tables and corresponding analyses were not shared with Smart Reporting GmbH.

2. Methods:

2.1 Submit a flow-chart of the SR and SOP structures and decision-trees for review (as supporting information).

A flow-chart of the SR and SOP structures and decision-trees was created and submitted for review as supporting information.

2.2 How many residents participated in preparing report? Were they at the same stage in their training? Did a given resident do an SR and a CR for the same patient? If so, would this introduce a bias for which ever report was prepared secondly? There are similarly questions if a given resident prepared reports on more than one patient.

In total, n=6 Otolaryngology residents with a mean work experience of 4.5 ± 0.9 years participated in preparing operation plannings. With respect to the low standard deviation, it can be assumed that they were at the same stage in their training. All structured radiology reports were created by one board-certified radiologist. These SRs were compared to pre-existing CRs which were created during clinical routine. 

Since participating residents used the same CT scans for corresponding SOPs and COPs, potential bias due to testing or learning effects cannot be ruled out. In a first step, COPs of the CT scans were created analogously to the clinical routine. In a second step, participating residents used the same images to generate corresponding SOPs. This sequence was chosen in order to reduce bias since, unlike SOPs, COPs do not offer any feedback to the user. Consequently, potential training effects are minimized to greatest possible extent. Additionally, residents prepared SOPs and COPs before the operation to reduce potential bias that may arise from additional knowledge acquired from intraoperative findings. 

Within this study, participating residents assigned to carry out FESS procedures under the direct supervision of the corresponding author one after another. Since there were only n=6 residents with a sufficient experience in FESS procedures available at the University Medical Center Mainz, each resident prepared 5 corresponding SOPs/COPs within this study. By assigning residents one after the other, potential bias due to learning effects cannot be ruled out but are minimized since the time interval between assignments is as long as possible. A long time interval may blur details used within SOPs.

These points raised by the reviewer were added and critically discussed in the discussion section of the revised manuscript.

3. Results:

Is there any data concerning an association or correlation between “completeness” (an outcome measure) and “treatment decisions on an individual level” (part of the overall aim of the study)?

Thank you for this important comment. We have to apologize that our statement concerning the overall aim of this study was potentially imprecise with regard to “treatment decisions on an individual level”. The principle aim of this study was to perform a comparative analysis of the completeness of CT reports and surgical plannings using SR and SOP, respectively, to the conventional approaches (e.g. CR and COP). Additionally, we hypothesized that an increase in completeness of radiology reports and operation plannings may have the potential to improve treatment decisions and patient outcome in the long term. This hypothesis has to be evaluated by future studies. 

4. Discussion:

4.1 How could it be determined whether the use of SR (for pre-operative CT images) and SOP “may decrease potential risks during FESS?

To evaluate the impact of structured reporting approaches on diagnosing and treating paranasal sinus disease, we plan to use three different approaches. In a first step, which is the topic of the present study, we evaluated structured reporting templates for radiology reporting and operation planning and their impact on report completeness. This study represents the preoperative part of the project. In a second step, we plan to evaluate the impact of a structured reporting template, which creates the actual operation report with respect to completeness, time efficiency and extent of the operation. Due to the structured and standardized format of the radiology report, the operation planning and the operation report, similarities and differences can be analyzed semi-automatically. Consequently, combining the preoperative and intraoperative approach could help determine whether the use of SR for pre-operative CT images and SOP may decrease potential risks during FESS by pointing out potential hazards before the operation. 

4.2 What about having a structured outcomes report (“SOR” for post-operative outcomes) to go along with the SR and SOP?

Thank you for this thoughtful comment that represents the final step of our project. By creating a structured reporting template for the documentation of postoperative care and postoperative outcome, the impact of the first two steps (as described in 4.1) can be evaluated with respect to the actual incidence of potential risks. Due to the pleasingly low rate of severe complications in FESS, a dedicated study will have to involve multiple centers in order to recruit a sufficient amount of patients thus generating the appropriate power.

Thank you very much in advance for re-reviewing our manuscript!

Best regards,

Dr. Benjamin Ernst

---

## [Decision Letter · Decision Letter 1]

10 Nov 2020

The role of structured reporting and structured operation planning in functional endoscopic sinus surgery

PONE-D-20-22030R1

Dear Dr. Ernst,

We’re pleased to inform you that your manuscript has been judged scientifically suitable for publication and will be formally accepted for publication once it meets all outstanding technical requirements.

Kind regards,

Giannicola Iannella, M.D

Academic Editor

PLOS ONE

Additional Editor Comments (optional):

the authors have well revised the manuscript in accordance with the comments of revision 2. in my opinion this is an interesting paper that is suitable to be published on PLOS-ONE. Congratulations on your efforts.

Best regards

Dr Giannicola Iannella

Reviewers' comments:

Reviewer #2: The authors provided appropriate and complete responses to all issues.

The following were noted:

(1) The detailed response to my review issue #1, regarding the potential for a conflict of interest, was comprehensive and fully adequate. Note an important point made by the authors, “Data were analyzed by employees of the University Medical Center Mainz with no relationship to Smart Reporting GmbH. Data tables and corresponding analyses were not shared with Smart Reporting GmbH.”

(2.1) Flow-charts of the SR and SOP structures were submitted as supporting information.

(2.2) The detailed response to my review issue #2.2 is fully adequate. Note that “In a first step, COPs of the CT scans were created analogously to the clinical routine. In a second step, participating residents used the same images to generate corresponding SOPs. This sequence was chosen in order to reduce bias since, unlike SOPs, COPs do not offer any feedback to the user.” Although there remains the possibility for bias, reasonable efforts have been made to reduce the possibility.

(4.2) In regard to the future possibility of a structured outcomes report, the authors noted the importance of this and “a dedicated study (for this) will have to involve multiple centers in order to recruit a sufficient amount of patients thus generating the appropriate power.”

---

## [Editor Report · Acceptance letter]

12 Nov 2020

PONE-D-20-22030R1 

The role of structured reporting and structured operation planning in functional endoscopic sinus surgery 

Dear Dr. Ernst:

I'm pleased to inform you that your manuscript has been deemed suitable for publication in PLOS ONE. Congratulations! Your manuscript is now with our production department. 

Kind regards, 

on behalf of

Dr. Giannicola Iannella 

Academic Editor

PLOS ONE